# Contrastive Active Inference

**Pietro Mazzaglia**
IDLab
Ghent University
pietro.mazzaglia@ugent.be

**Tim Verbelen**
IDLab
Ghent University
tim.verbelen@ugent.be

**Bart Dhoedt**
IDLab
Ghent University
bart.dhoedt@ugent.be

## Abstract

Active inference is a unifying theory for perception and action resting upon the idea that the brain maintains an internal model of the world by minimizing free energy. From a behavioral perspective, active inference agents can be seen as self-evidencing beings that act to fulfill their optimistic predictions, namely preferred outcomes or goals. In contrast, reinforcement learning requires human-designed rewards to accomplish any desired outcome. Although active inference could provide a more natural self-supervised objective for control, its applicability has been limited because of the shortcomings in scaling the approach to complex environments. In this work, we propose a contrastive objective for active inference that strongly reduces the computational burden in learning the agent's generative model and planning future actions. Our method performs notably better than likelihood-based active inference in image-based tasks, while also being computationally cheaper and easier to train. We compare to reinforcement learning agents that have access to human-designed reward functions, showing that our approach closely matches their performance. Finally, we also show that contrastive methods perform significantly better in the case of distractors in the environment and that our method is able to generalize goals to variations in the background.

## 1 Introduction

Deep Reinforcement Learning (RL) has led to successful results in several domains, such as robotics, video games and board games [42, 36, 2]. From a neuroscience perspective, the reward prediction error signal that drives learning in deep RL closely relates to the neural activity of dopamine neurons for reward-based learning [44, 3]. However, the reward functions used in deep RL typically require domain and task-specific design from humans, spoiling the generalization capabilities of RL agents. Furthermore, the possibility of faulty reward functions makes the application of deep RL risky in real-world contexts, given the possible unexpected behaviors that may derive from it [10, 29, 38].

Active Inference (AIF) has recently emerged as a unifying framework for learning perception and action. In AIF, agents operate according to one absolute imperative: minimize their free energy [15]. With respect to past experience, this encourages to update an internal model of the world to maximize evidence with respect to sensory data. With regard to future actions, the inference process becomes 'active' and agents select behaviors that fulfill optimistic predictions of their model, which are represented as preferred outcomes or goals [17]. Compared to RL, the AIF framework provides a more natural way of encoding objectives for control. However, its applicability has been limited because of the shortcomings in scaling the approach to complex environments, and current implementations have focused on tasks with either low-dimensional sensory inputs and/or small sets of discrete actions [12]. Moreover, several experiments in the literature have replaced the agent's preferred outcomes with RL-like rewards from the environment, downplaying the AIF potential to provide self-supervised objectives [13, 34, 49].

35th Conference on Neural Information Processing Systems (NeurIPS 2021).

One of the major shortcomings in scaling AIF to environments with high-dimensional, e.g. image-based, environments comes from the necessity of building accurate models of the world, which try to reconstruct every detail in the sensory data. This complexity is also reflected in the control stage, when AIF agents compare future imaginary outcomes of potential actions with their goals, to select the most convenient behaviors. In particular, we advocate that fulfilling goals in image space can be poorly informative to build an objective for control.

In this work, we propose Contrastive Active Inference, a framework for AIF that aims to both reduce the complexity of the agent's internal model and to propose a more suitable objective to fulfill preferred outcomes, by exploiting contrastive learning. Our method provides a self-supervised objective that constantly informs the agent about the distance from its goal, without needing to reconstruct the outputs of potential actions in high-dimensional image space.

The contributions of our work can be summarised as follows: (*i*) we propose a framework for AIF that drastically reduces the computational power required both for learning the model and planning future actions, (*ii*) we combine our method with value iteration methods for planning, inspired by the RL literature, to amortize the cost of planning in AIF, (*iii*) we compare our framework to state-of-the-art RL techniques and to a non-contrastive AIF formulation, showing that our method compares well with reward-based systems and outperforms non-contrastive AIF, (*iv*) we show that contrastive methods work better than reconstruction-based methods in presence of distractors in the environment, (*v*) we found that our contrastive objective for control allows matching desired goals, despite differences in the backgrounds. The latter finding could have important consequences for deploying AIF in real-world settings, such as robotics, where perfectly reconstructing observations from the environment and matching them with high-dimensional preferences is practically unfeasible.

## 2  Background

The control setting can be formalized as a Partially Observable Markov Decision Process (POMDP), which is denoted with the tuple $\mathcal{M} = \{\mathcal{S}, \mathcal{A}, T, \Omega, \mathcal{O}, \gamma\}$, where $\mathcal{S}$ is the set of unobserved states, $\mathcal{A}$ is the set of actions, $T$ is the state transition function, also referred to as the dynamics of the environment, $\Omega$ is the set observations, $\mathcal{O}$ is a set of conditional observation probabilities, and $\gamma$ is a discount factor (Figure 1). We use the terms observations and outcomes interchangeably throughout the work. In RL, the agent has also access to a reward function $R$, mapping state-action pairs to rewards.

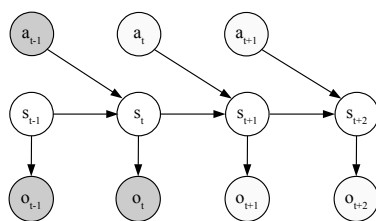

Figure 1: *POMDP Graphical Model*

**Active Inference.** In AIF, the goal of the agent is to minimize (a variational bound on) the surprisal over observations, $-\log p(o)$. With respect to past observations, the upper bound leads to the variational free energy $\mathcal{F}$, which for timestep $t$ is:

$$\mathcal{F} = E_{q(s_t)}\left[\log q(s_t) - \log p(o_t, s_t)\right] \geq -\log p(o_t) \tag{1}$$

where $q(s_t)$ represents an approximate posterior.

The agent hence builds a generative model over states, actions and observations, by defining a state transition function $p(s_t|s_{t-1}, a_{t-1})$ and a likelihood mapping $p(o_t|s_t)$, while the posterior distribution over states is approximated by the variational distribution $q(s_t|o_t)$. The free energy can then be decomposed as:

$$\mathcal{F}_{\text{AIF}} = \underbrace{D_{\text{KL}}\left[q(s_t|o_t)||p(s_t|s_{t-1}, a_{t-1})\right]}_{\text{complexity}} - \underbrace{E_{q(s_t|o_t)}[\log p(o_t|s_t)]}_{\text{accuracy}}. \tag{2}$$

This implies that minimizing variational free energy, on the one hand, maximizes the likelihood of observations under the likelihood mapping (i.e. maximizing accuracy), whilst minimizing the KL divergence between the approximate posterior and prior (i.e. complexity). Note that for the past we assume that outcomes and actions are observed, hence only inferences are made about the state $s_t$. Also note that the variational free energy is defined as the negative evidence lower bound as known from the variational autoencoder framework [39, 27].

For future timesteps, the agent has to make inferences about both future states and actions $q(s_t, a_t) = q(a_t|s_t)q(s_t)$, while taking into account expectations over future observations. Crucially, in active

inference the agent has a prior distribution $\tilde{p}(o_t)$ on preferred outcomes it expects to obtain. Action selection is then cast as an inference problem, i.e. inferring actions that will yield preferred outcomes, or more formally that minimize the expected free energy $\mathcal{G}$:

$$\mathcal{G} = E_{q(o_t, s_t, a_t)} \left[ \log q(s_t, a_t) - \log \tilde{p}(o_t, s_t, a_t) \right], \tag{3}$$

where $\tilde{p}(o_t, s_t, a_t) = p(a_t) p(s_t | o_t) \tilde{p}(o_t)$ is the agent's biased generative model, and the expectation is over predicted observations, states and actions $q(o_t, s_t, a_t) = p(o_t | s_t) q(s_t, a_t)$.

If we assume the variational posterior over states is a good approximation of the true posterior, i.e. $q(s_t | o_t) \approx p(s_t | o_t)$, and we also consider a uniform prior $p(a_t)$ over actions [35], the expected free energy can be formulated as:

$$\mathcal{G}_{\text{AIF}} = - \underbrace{E_{q(o_t)}[D_{\text{KL}}\left[q(s_t | o_t) || q(s_t)\right]]}_{\text{intrinsic value}} - \underbrace{E_{q(o_t)}[\log \tilde{p}(o_t)]}_{\text{extrinsic value}} - \underbrace{E_{q(s_t)}[\mathcal{H}(q(a_t | s_t))]}_{\text{action entropy}}. \tag{4}$$

Intuitively, this means that the agent will infer actions for which observations have a high information gain about the states (i.e. intrinsic value), which will yield preferred outcomes (i.e. extrinsic value), while also keeping its possible actions as varied as possible (i.e. action entropy).

Full derivations of the equations in this section are provided in the Appendix.

**Reinforcement Learning.** In RL, the objective of the agent is to maximize the discounted sum of rewards, or return, over time $\sum_t^\infty \gamma^t r_t$. Deep RL can also be cast as probabilistic inference, by introducing an optimality variable $\mathcal{O}_t$ which denotes whether the time step $t$ is optimal [30]. The distribution over the optimality variable is defined in terms of rewards as $p(\mathcal{O}_t = 1 | s_t, a_t) = \exp(r(s_t, a_t))$. Inference is then obtained by optimizing the following variational lower bound

$$-\log p(\mathcal{O}_t) \le E_{q(s_t, a_t)} \left[ \log q(s_t, a_t) - \log p(\mathcal{O}_t, s_t, a_t) \right]$$
$$= -E_{q(s_t, a_t)}[r(s_t, a_t)] - E_{q(s_t)}[\mathcal{H}(q(a_t | s_t))], \tag{5}$$

where the reward-maximizing RL objective is augmented with an action entropy term, as in maximum entropy control [20]. As also highlighted in [35], if we assume $\log \tilde{p}(o_t | s_t) = \log p(\mathcal{O}_t | s_t)$, we can see that RL works alike AIF, but encoding optimality value in the likelihood rather than in the prior.

In order to improve sample-efficiency of RL, model-based approaches (MBRL), where the agent relies on an internal model of the environment to plan high-rewarding actions, have been studied.

**Contrastive Learning.** Contrastive representations, which aim to organize the data distinguishing similar and dissimilar pairs, can be learned through Noise Contrastive Estimation (NCE) [19]. Following [37], an NCE loss can be defined as a lower bound on the mutual information between two variables. Given two random variables $X$ and $Y$, the NCE lower bound is:

$$I(X; Y) \ge I_{\text{NCE}}(X; Y) \triangleq \mathbb{E} \left[ \frac{1}{K} \sum_{i=1}^{K} \log \frac{e^{f(x_i, y_i)}}{\frac{1}{K} \sum_{j=1}^{K} e^{f(x_i, y_j)}} \right], \tag{6}$$

where the expectation is over $K$ independent samples from the joint distribution: $\prod_j p(x_j, y_j)$ and $f(x, y)$ is a function, called critic, that approximates the log density ratio $\log \frac{p(x|y)}{p(x)}$. Crucially, the critic can be unbounded, as in [50], where the authors showed that an inner product of transformated samples from X and Y, namely $f(x, y) = h(x)^T g(y)$, with $h$ and $g$ functions, works well as a critic.

## 3 Contrastive Active Inference

In this section, we present the Contrastive Active Inference framework, which reformulates the problem of optimizing the free energy of the past $\mathcal{F}$ and the expected free energy of the future $\mathcal{G}$ as contrastive learning problems.

### 3.1 Contrastive Free Energy of the Past

In order to learn a generative model of the environment following AIF, an agent could minimize the variational free energy $\mathcal{F}_{\text{AIF}}$ from Equation 2. For high-dimensional signals, such as pixel-based

images, the model works similarly to a Variational AutoEncoder (VAE) [27], with the information encoded in the latent state $s_t$ being used to produce reconstructions of the high-dimensional observations $o_t$ through the likelihood model.

However, reconstructing images at pixel level has several shortfalls: (*a*) it requires models with high capacity, (*b*) it can be quite computationally expensive, and (*c*) there is the risk that most of the representation capacity is wasted on complex details of the images that are irrelevant for the task.

We can avoid predicting observations, by using an NCE loss. Optimizing the mutual information between states and observations, it becomes possible to infer $s_t$ from $o_t$, without having to compute a reconstruction. In order to turn the variational free energy loss into a contrastive loss, we add the constant marginal log-probability of the data $\log p(o_t)$ to $\mathcal{F}$, obtaining:

$$
\begin{aligned}
\mathcal{F} &\stackrel{+}{=} D_{\mathrm{KL}}\left[q(s_t|o_t)\|p(s_t)\right] - E_{q(s_t|o_t)}[\log p(o_t|s_t) - \log p(o_t)] \\
&= D_{\mathrm{KL}}\left[q(s_t|o_t)\|p(s_t)\right] - I(S_t; O_t).
\end{aligned}
\tag{7}
$$

As for Equation 6, we can apply a lower bound on the mutual information $I(S_t; O_t)$. We can define the contrastive free energy of the past as:

$$
\begin{aligned}
\mathcal{F}_{\mathrm{NCE}} &= D_{\mathrm{KL}}\left[q(s_t|o_t)\|p(s_t)\right] - I_{\mathrm{NCE}}(S_t; O_t) \\
&= D_{\mathrm{KL}}\left[q(s_t|o_t)\|p(s_t)\right] - \mathbb{E}_{q(s_t|o_t)p(o_t)}[f(o_t, s_t)] + \mathbb{E}_{q(s_t|o_t)p(o')}[\log \tfrac{1}{K} \textstyle\sum_{j=1}^{K} e^{f(o_j, s_t)}],
\end{aligned}
\tag{8}
$$

where the dynamics $p(s_t)$ is modelled as $p(s_t|s_{t-1}, a_{t-1})$, and the $K$ samples from the distribution $p(o')$ represent observations that do not match with the state $s_t$, catalyzing the contrastive mechanism. Given the inequality $I_{\mathrm{NCE}} \leq I$, this contrastive utility provides an upper bound on the variational free energy, $\mathcal{F} \leq \mathcal{F}_{\mathrm{NCE}}$, and thus on suprisal.

## 3.2 Contrastive Free Energy of the Future

Performing active inference for action selection means inferring actions that realize preferred outcomes, by minimizing the expected free energy $\mathcal{G}$. In order to assess how likely expected future outcomes are to fulfill the agent's preferences, in Equation 4, the agent uses its generative model to predict future observations.

Reconstructing imaginary observations in the future can be computationally expensive. Furthermore, matching imagined outcomes with the agent's preferences in pixel space can be poorly informative, as pixels are not supposed to capture any semantics about observations. Also, observations that are "far" in pixel space aren't necessarily far in transition space. For example, when the goal is behind a door, standing before the door is "far" in pixel space but only one action away (i.e. opening the door).

When the agent learns a contrastive model of the world, following Equation 8, it can exploit its ability to match observations with states without reconstructions, in order to search for the states that correspond with its preferences. Hence, we formulate the expectation in the expected free energy $\mathcal{G}$ in terms of the preferred outcomes, so that we can add the constant marginal $\tilde{p}(o_t)$, obtaining:

$$
\begin{aligned}
\mathcal{G} &\stackrel{+}{=} E_{\tilde{p}(o_t)q(s_t, a_t)}\left[\log q(s_t, a_t) - \log \tilde{p}(o_t, s_t, a_t) + \log \tilde{p}(o_t)\right] \\
&= D_{\mathrm{KL}}\left[q(s_t)\|p(s_t)\right] - I(S_t; \tilde{O}_t) - E_{q(s_t)}[\mathcal{H}(q(a_t|s_t))].
\end{aligned}
\tag{9}
$$

With abuse of notation, the mutual information between $S_t$ and $\tilde{O}_t$ quantifies the amount of information shared between future imaginary states and preferred outcomes.

We further assume $D_{\mathrm{KL}}\left[q(s_t)\|p(s_t)\right] = 0$, which constrains the agent to only modify its actions, preventing it to change the dynamics of the world to accomplish its goal, as pointed out in [30]. This leads to the following objective for the contrastive free energy of the future:

$$
\begin{aligned}
\mathcal{G}_{\mathrm{NCE}} &= -I_{\mathrm{NCE}}(S_t; \tilde{O}_t) - E_{q(s_t)}[\mathcal{H}(q(a_t|s_t))] \\
&= -\mathbb{E}_{q(s_t)\tilde{p}(o)}[f(\tilde{o}, s_t)] + \mathbb{E}_{q(s_t)p(o')}[\log \tfrac{1}{K} \textstyle\sum_{j=1}^{K} e^{f(o_j, s_t)}] - E_{q(s_t)}[\mathcal{H}(q(a_t|s_t))].
\end{aligned}
\tag{10}
$$

Similar as in the $\mathcal{F}_{\mathrm{NCE}}$, the $K$ samples from $p(o')$ foster the contrastive mechanism, ensuring that the state $s_t$ corresponds to the preferred outcomes, while also being as distinguishable as possible from other observations. This component implies a similar process as to the ambiguity minimization aspect typically associated with the AIF framework [16].

# 4 Model and Algorithm

The AIF framework entails perception and action, in a unified view. In practice, this is translated into learning a world model, to capture the underlying dynamics of the environment, minimizing the free energy of the past, and learning a behavior model, which proposes actions to accomplish the agent's preferences, minimizing the free energy of the future. In this work, we exploit the high expressiveness of deep neural networks to learn the world and the behavior models.

The world model is composed by the following components:

| | |
|---|---|
| Prior network: | $p_\phi(s_t|s_{t-1}, a_{t-1})$ |
| Posterior network: | $q_\phi(s_t|s_{t-1}, a_{t-1}, o_t)$ |
| Representation model: | $f_\phi(o, s)$ |

For the prior network, we use a GRU [9] while the posterior network combines a GRU with a CNN to process observations. Both the prior and the posterior outputs are used to parameterize Gaussian multivariate distributions, which represent a stochastic state, from which we sample using the reparameterization trick [27]. This setup is inspired upon the models presented in [21, 54, 4]. For the representation model, we utilize a network that first processes $o_t$ and $s_t$ with MLPs and then computes the dot-product between the outputs, obtaining $f_\phi(o, s) = h_\phi(o)^T g_\phi(s)$, analogously to [50]. We indicate the unified world model loss with: $J_\phi = \sum_t \mathcal{F}_{\text{NCE}}(s_t, o_t)$.

In order to amortize the cost of long-term planning for behavior learning, we use an expected utility function $g(s_t)$ to estimate the expected free energy in the future for the state $s_t$, similarly to [34]. The behavior model is then composed by the following components:

| | |
|---|---|
| Action network: | $q_\theta(a_t|s_t)$ |
| Expected utility network: | $g_\psi(s_t)$ |

where the action and expected utility networks are both MLPs that are concurrently trained as in actor-critic architectures for RL [28, 20]. The action model aims to minimize the expected utility, which is an estimate of the expected free energy of the future over a potentially infinite horizon, while the utility network aims to predict a good estimate of the expected free energy of the future that is obtainable by following the actions of the action network. We indicate the action network loss with $J_\theta = \sum_t \mathcal{G}_{\text{NCE}}(s_t)$ and the utility network loss with $J_\psi = \sum_t (g_\psi(s_t) - \sum_{k=T}^\infty \mathcal{G}_{\text{NCE}}(s_t))^2$, where the sum from the current time step to an infinite horizon is obtained by using a TD($\lambda$) exponentially-weighted estimator that trades off bias and variance [43] (details in Appendix).

The training routine, which alternates updates to the models with data collection, is shown in Algorithm 1. At each training iteration of the model, we sample $B$ trajectories of length $L$ from the replay buffer $D$. Negative samples for the contrastive functionals are selected, for each state, by taking $L - 1$ intra-episode negatives, corresponding to temporally different observations, and $(B - 1) * L$ extra-episode negatives, corresponding to observations from different episodes.

Most of the above choices, along with the training routine itself, are deliberately inspired to current state-of-the-art approaches for MBRL [23, 22, 11]. The motivation behind this is twofold: on the one hand, we want to show that approaches that have been used to scale RL for complex planning can also straightforwardly be applied for scaling AIF. On the other hand, in the next section, we offer a direct comparison to current state-of-the-art techniques for RL that, being unbiased with respect to the models' architecture and the training routine, can focus on the relevant contributions of this paper, which concerns the contrastive functionals for perception and action.

Relevant parameterization for the experiments can be found in the next section, while hyperparameters and a detailed description of each network are left to the Appendix.

# 5 Experiments

In this section, we compare the contrastive AIF method to likelihood-based AIF and MBRL in high-dimensional image-based settings. As the experiments are based in environments originally designed for RL, we defined ad-hoc preferred outcomes for AIF. Our experimentation aims to answer the following questions: (i) is it possible to achieve high-dimensional goals with AIF-based methods? (ii) what is the difference in performance between RL-based and AIF-based methods? (iii) does

**Algorithm 1:** Training and Data Collection
________________________________________________________

1:  Initialize world model parameters $\phi$ and behavior model parameters $\theta$ and $\psi$.
2:  Initialize dataset $\mathcal{D}$ with R random-action episodes.
3:  **while** *not done* **do**
4:      **for** *update step u=1..U* **do**
5:          Sample B trajectories of length L from $\mathcal{D}$.
6:          Infer states $s_t$ using the world model.
7:          Update the world model parameters $\phi$ on the B trajectories, minimizing $\mathcal{L}_\theta$.
8:          Imagine I trajectories of length H from each $s_t$.
9:          Update the action network parameters $\theta$ on the I trajectories, minimizing $\mathcal{L}_\phi$.
10:         Update the expected utility network parameters $\psi$ on the I trajectories, minimizing $\mathcal{L}_\psi$.
11:     **end**
12:     Reset the environment.
13:     Init state $s_t = s_0$ and set t = 0
14:     Init new trajectory with the first observation $\mathcal{T} = \{o_1\}$
15:     **while** *environment not done* **do**
16:         Infer action $a_t$ using the action network $q_\theta(a_t|s_t)$.
17:         Act on the environment with $a_t$, and receive observation $o_{t+1}$.
18:         Add transition to the buffer $\mathcal{T} = \mathcal{T} \cup \{a_t, o_{t+1}\}$ and set t = t + 1
19:         Infer state $s_t$ using the world model
20:     **end**
21: **end**
________________________________________________________

contrastive AIF perform better than likelihood-based AIF? (iv) in what contexts contrastive methods are more desirable than likelihood-based methods? (v) are AIF-based methods resilient to variations in the environment background?

We compare the following four flavors of MBRL and AIF, sharing similar model architectures and all trained according to Algorithm 1:

- *Dreamer*: the agents build a world model able to reconstruct both observations and rewards from the state. Reconstructed rewards for imagined trajectories are then used to optimize the behavior model in an MBRL fashion [23, 22].
- *Contrastive Dreamer*: this method is analog to its reconstruction-based counterpart, apart from that it uses a contrastive representation model, like our approach. Similar methods have been studied in [23, 32].
- *Likelihood-AIF*: the agent minimizes the AIF functionals, using observation reconstructions. The representation model from the previous section is replaced with an observation likelihood model $p_\phi(o_t|s_t)$, which we model as a transposed CNN. Similar approaches have been presented in [13, 34].
- *Contrastive-AIF* (**ours**): the agent minimizes the contrastive free energy functionals.

In Table 1, we compare the number of parameters and of multiply-accumulate (MAC) operations required for the two flavors of the representation model in our implementation: likelihood-based and contrastive (ours). Using a contrastive representation makes the model 13.8 times more efficient in terms of MAC operations and reduces the number of parameters by a factor 3.5.

In Table 2, we compare the computation speed in our experiments, measuring wall-clock time and using Dreamer as a reference. Contrastive methods are on average 16% faster, while Likelihood-AIF, which in addition to Dreamer reconstructs observations for behavior learning, is 224% slower.



Table 1: Computational Efficiency

|            | MMACs | # Params |
|------------|-------|----------|
| Likelihood | 212.2 | 4485.7k  |
| Ours       | 15.4  | 1266.7k  |

Table 2: Computation Time

|                         | w.r.t. Dreamer |
|-------------------------|----------------|
| Contrastive Dreamer/AIF | 0.84           |
| Likelihood-AIF          | 3.24           |



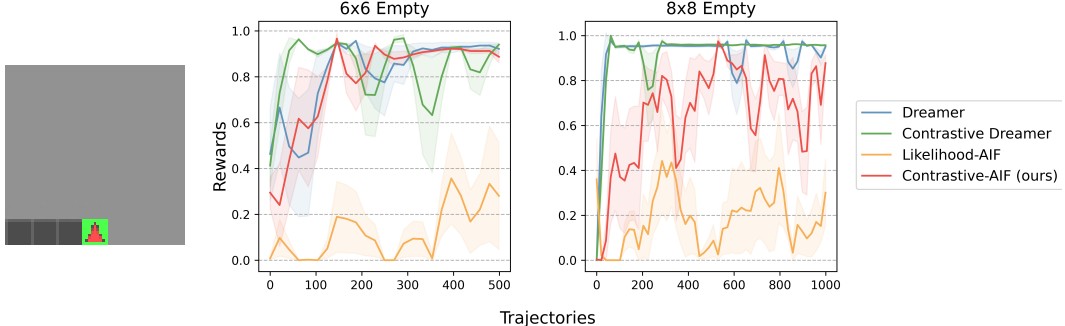

Figure 2: **MiniGrid Experiments.** (left) Empty task goal image. (right) Results: shaded areas indicate standard deviation across several runs.

## 5.1 MiniGrid Navigation

We performed experiments on the Empty 6×6 and the Empty 8×8 environments from the MiniGrid suite [8]. In these tasks, the agent, represented as a red arrow, should reach the goal green square navigating a black grid (see Figure 3a). The agent only sees a part of the environment, corresponding to a 7×7 grid centered on the agent (in the bottom center tile). We render observations as 64×64 pixels. For RL, a positive reward between 0 and 1 is provided to the agent as soon as the agent reaches the goal tile: the faster the agent reaches the goal, the higher the reward. For AIF agents, we defined the preferred outcome as the agent seeing itself on the goal green tile, as shown in Figure 2 (left).

For the 6×6 task, the world model is trained by sampling $B = 50$ trajectories of length $L = 7$, while the behavior model is trained by imagining $H = 6$ steps long trajectories. For the 8×8 task, we increased the length $L$ to 11 and the imagination horizon $H$ to 10. For both tasks, we first collected $R = 50$ random episodes, to populate the replay buffer, and train for $U = 100$ steps after collecting a new trajectory. Being the action set discrete, we optimized the action network employing REINFORCE gradients [52] with respect to the expected utility network's estimates.

We assess performance in terms of the rewards achieved along one trajectory, stressing that AIF methods did not have access to the reward function but only to the goal observation, during training. The results, displayed in Figure 2 (right), show the average sum of rewards obtained along training, over the number of trajectories collected. We chose to compare over the number of trajectories as the trajectories' length depends on whether the agent completed the task or not.

In this benchmark, we see that MBRL algorithms rapidly converge to highly rewarding trajectories, in both the 6×6 and the 8×8 tasks. Likelihood-AIF struggles to converge to trajectories that reach the goal consistently and fast, mostly achieving a reward mean lower than 0.4. In contrast, our method performs comparably to the MBRL methods in the 6×6 grid and reaches the goal twice more consistently than Likelihood-AIF in the 8×8 grid, leaning towards Dreamer and Contrastive Dreamer's results.

**Utility Function Analysis.** In order to understand the differences between the utility functions we experimented with, we analyze the values assigned to each tile in the 8×8 task by every method.

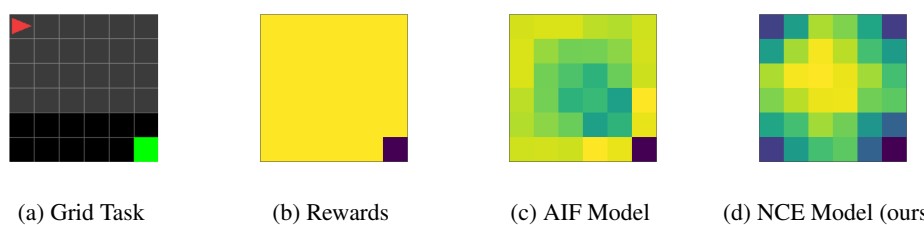

| (a) Grid Task | (b) Rewards | (c) AIF Model | (d) NCE Model (ours) |

Figure 3: **Utility Values MiniGrid.** (b-c-d) Darker tiles correspond to higher utility values.

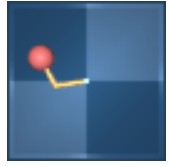
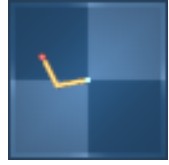
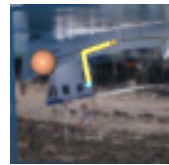

(a) Reacher Easy Goal        (b) Reacher Hard Goal        (c) Distracting Reacher Easy

Figure 4: **Continuous tasks setup.** Note that the Reacher Easy Goal is also used for the Distracting Reacher Easy task, without changing the goal's background.

For the AIF methods, we collected all possible transitions in the environment and used the model to compute utility values for each tile. The results are shown in Figure 3.

The reward signal for the Empty environment is very sparse and informative only once the agent reaches the goal. In contrast, AIF methods provide denser utility values. In particular, we noticed that the Likelihood-AIF model provides a very strong signal for the goal position, whereas other values are less informative of the goal. Instead, the Contrastive-AIF model seems to capture some semantic information about the environment: it assigns high values to all corners, which are conceptually closer outcomes to the goal, while also providing the steepest signal for the green corner and its neighbor tiles. As also supported by the results obtained in terms of rewards, our method provides a denser and more informative signal to reach the goal in this task.

## 5.2 Reacher Task

We performed continuous-control experiments on the Reacher Easy and Hard tasks from the Deep-Mind Control (DMC) Suite [48] and on Reacher Easy from the Distracting Control Suite [47]. In this task, a two-link arm should penetrate a goal sphere with its tip in order to obtain rewards, with the sphere being bigger in the Easy task and smaller in the Hard one. The Distracting Suite adds an extra layer of complexity to the environment, altering the camera angle, the arm and the goal colors, and the background. In particular, we used the 'easy' version of this benchmark, corresponding to smaller changes in the camera angles and in the colors, and choosing the background from one of four videos (example in Figure 4c).

In order to provide consistent goals for the AIF agents, we fixed the goal sphere position as shown in Figure 4b and 4a. As there is no fixed background in the Distracting Suite task, we could not use a goal image with the correct background, as that would have meant changing it at every trajectory. To not introduce 'external' interventions into the AIF experiments, we decided to use a goal image with the original blue background from the DMC Suite to test out the AIF capability to generalize goals to environments having the same dynamics but different backgrounds.

For both tasks, the world model is trained by sampling $B = 30$ trajectories of length $L = 30$, while the behavior model is trained by imagining $H = 10$ steps long trajectories. We first collect $R = 50$ random episodes, to populate the replay buffer, and train for $U = 100$ steps after every new trajectory. Being the action set continuous, we optimized the action network backpropagating the expected utility value through the dynamics, by using the reparameterization trick for sampling actions [23, 11].

The results are presented in Figure 5, evaluating agents in term of the rewards obtained per trajectory. The length of a trajectory is fixed to $1 \cdot 10^3$ steps.

**Reacher Easy/Hard.** The results on the Reacher Easy and Hard tasks show that our method was the fastest to converge to stable high rewards, with Contrastive Dreamer and Dreamer following. In particular, Dreamer's delay to convergence should be associated with the more complex model, that took more epochs of training than the contrastive ones to provide good imagined trajectories for planning, especially for the Hard task. The Likelihood-AIF failed to converge in all runs, because of the difficulty of matching the goal state in pixel space, which only differs a small number of pixels from any other environment observation.

**Distracting Reacher Easy.** On the Distracting task, we found that Dreamer failed to succeed. As we show in Appendix, the reconstruction model's capacity was entirely spent on reconstructing the complex backgrounds, failing to capture relevant information for the task. Conversely, Contrastive Dreamer was able to ignore the complexity of the observations and the distractions present in the

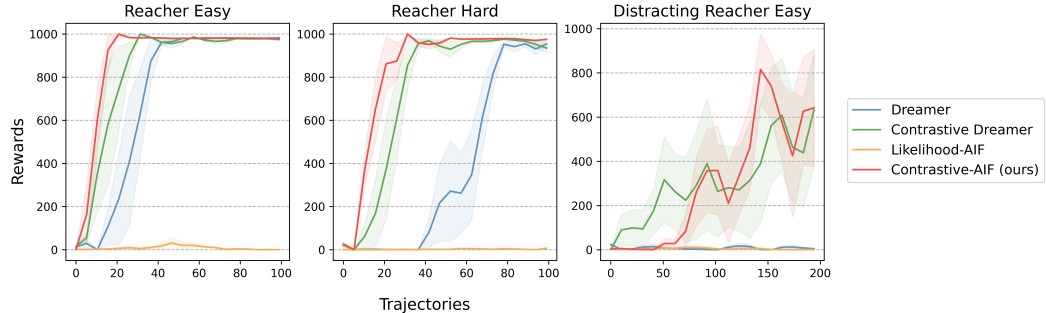

Figure 5: **Reacher Results.** Shaded areas indicate standard deviation across several runs.

environment, eventually succeeding in the task. Surprisingly, also our Contrastive-AIF method was able to succeed, showing generalization capabilities that are not shared by the likelihood counterpart.

We believe this result is important for two reasons: (1) it provides evidence that contrastive features better capture semantic information in the environment, potentially ignoring complex irrelevant details, (2) contrastive objectives for planning can be invariant to changes in the background, when the underlying dynamics of the task stays the same.

**Utility Function Analysis.** To collect further insights on the different methods' objectives, we analyze the utility values assigned to observations with different poses in the Reacher Hard task. In Figure 6, we show a comparison where all the values are normalized in the range [0,1], considering the maximum and minimum values achievable by each method.

The reward signal is sparse and provided only when the arm is penetrating the goal sphere with his orange tip. In particular, a reward of +1 is obtained only when the tip is entirely contained in the sphere. The Likelihood-AIF utility looks very flat due to the static background, which causes any observation to be very similar to the preferred outcome in pixel space. Even a pose that is very different from the goal, such as the top left one, is separated only by a relatively small number of pixels from the goal one, in the bottom right corner, and this translates into very minor differences in utility values (i.e. 0.98 vs 1.00). For Contrastive-AIF, we see that the model provides higher utility values for observations that look perceptually similar to the goal and lower values for more distant states, providing a denser signal to optimize for reaching the goal. This was certainly crucial in achieving the task in this experiment, though overly-shaped utility functions can be more difficult to optimize [1], and future work should analyze the consequences of such dense shaping.

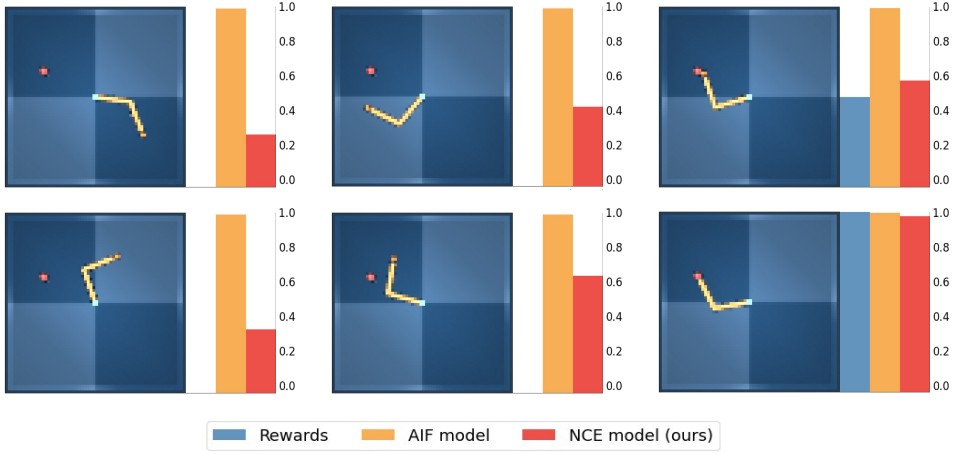

Figure 6: **Utility Values Reacher.** Normalized utility values for multiple poses in Reacher Hard.

# 6 Related Work

**Contrastive Learning.** Contrastive learning methods have recently led to important breakthroughs in the unsupervised learning landscape. Techniques like MoCO [7, 24] and SimCLR [5, 6] have progressively improved performance in image recognition, by using only a few supervised labels. Contrastive learning representations have also shown successful when employed for natural language processing [50] and model-free RL [46].

**Model-based Control.** Improvements in the dynamics generative model [21], have recently allowed model-based RL methods to reach state-of-the-art performance, both in control tasks [23] and on video games [22, 26]. An important line of research focuses on correctly balancing real-world experience with data generated from the internal model of the agent [25, 11].

**Outcome-Driven Control.** The idea of using desired outcomes to generate control objectives has been explored in RL as well [41, 18, 40]. In [31], the authors propose a system that, given a desired goal, can sample plans of action from a latent space and decode them to act on the environment. DISCERN [51] maximizes mutual information to the goal, using cosine similarity between the goal and a given observation, in the feature space of a CNN model.

**Active Inference.** In our work, we used active inference to derive actions, which is just one possibility to perform AIF, as discussed in [14, 35]. In other works, the expected free energy is passively used as the utility function to select the best behavior among potential sequences of actions [15, 16]. Methods that combine the expressiveness of neural networks with AIF have been raising in popularity in the last years [53]. In [13], the authors propose an amortized version of Monte Carlo Tree Search, through an habit network, for planning. In [49], AIF is seen performing better than RL algorithms in terms of reward maximization and exploration, on small-scale tasks. In [34], they propose an objective to amortize planning in a value iteration fashion.

# 7 Discussion

We presented the Contrastive Active Inference framework, a contrastive learning approach for active inference, that casts the free energy minimization imperatives of AIF as contrastive learning problems. We derived the contrastive objective functionals and we corroborated their applicability through empirical experimentation, in both continuous and discrete action settings, with high-dimensional observations. Combining our method with models and learning routines inspired from the model-based RL scene, we found that our approach can perform comparably to models that have access to human-designed rewards. Our results show that contrastive features better capture relevant information about the dynamics of the task, which can be exploited both to find conceptually similar states to preferred outcomes and to make the agent's preferences invariant to irrelevant changes in the environment (e.g. background, colors, camera angle).

While the possibility to match states to outcomes in terms of similar features is rather convenient in image-based tasks, the risk is that, if the agent never saw the desired outcome, it would converge to the semantically closest state in the environment that it knows. This raises important concerns about the necessity to provide good exploratory data about the environment, in order to prevent the agent from hanging in local minima. For this reason, we aim to look into combining our agent with exploration-driven data collection, for zero-shot goal achievement [33, 45]. Another complementary line of research would be equipping our method with better experience replay mechanisms, such as HER [1], to improve the generalization capabilities of the system.

## Broader impact

Active inference is a biologically-plausible unifying theory for perception and action. Implementations of active inference that are both tractable and computationally cheap are important to foster further research towards potentially better theories of the human brain. By strongly reducing the computational requirements of our system, compared to other deep active inference implementations, we aim to make the study of this framework more accessible. Furthermore, our successful results on the robotic manipulator task with varying realistic backgrounds show that contrastive methods are promising for real-world applications with complex observations and distracting elements.

## Acknowledgments and Disclosure of Funding

This research received funding from the Flemish Government (AI Research Program).

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
