# A  Background Derivations

In this section, we provide the derivations of the equations provided in section 2.

In all equations, both for the past and the future, we consider only one time step $t$. This is possible thanks to the Markov assumption, stating that the environment properties exclusively depend on the previous time step. This makes possible to write step-wise formulas, by applying ancestral sampling, i.e. for the state dynamics until $T$:

$$\log p(s_{\leq T}|a_{\leq T}) = \sum_{t=1}^{T} \log p(s_t|s_{t-1}, a_{t-1}).$$

To simplify and shorten the Equations, we mostly omit conditioning on past states and actions. However, as shown in section 4, the transition dynamics explicitly take ancestral sampling into account, by using recurrent neural networks that process multiple time steps.

## A.1  Free Energy of the Past

For past observations, the objective is to build a model of the environment for perception. Since computing the posterior $p(s_t|o_t)$ is intractable, we learn to approximate it with a variational distribution $q(s_t)$. As we show, this process provides an upper bound on the surprisal (log evidence) of the model:

$$\begin{aligned}
-\log p(o_t) &= -\log \sum_{s_t} p(o_t, s_t) \\
&= -\log \sum_{s_t} \frac{p(o_t, s_t)q(s_t)}{q(s_t)} \\
&= -\log \left[ E_{q(s_t)} \left[ \frac{p(o_t, s_t)}{q(s_t)} \right] \right] \\
&\leq -E_{q(s_t)} \left[ \log \frac{p(o_t, s_t)}{q(s_t)} \right] \\
&= E_{q(s_t)} \left[ \log q(s_t) - \log p(o_t, s_t) \right],
\end{aligned}$$

where we applied Jensen's inequality in the fourth row, obtaining the variational free energy $\mathcal{F}$ (Equation 1).

The free energy of the past can be mainly rewritten in two ways:

$$\begin{aligned}
\mathcal{F} &= E_{q(s_t)} \left[ \log q(s_t) - \log p(o_t, s_t) \right] \\
&= \underbrace{D_{\mathrm{KL}} \left[ q(s_t)||p(s_t|o_t) \right]}_{\text{evidence bound}} - \underbrace{\log p(o_t)}_{\text{log evidence}}. \\
&= \underbrace{D_{\mathrm{KL}} \left[ q(s_t)||p(s_t) \right]}_{\text{complexity}} - \underbrace{E_{q(s_t)} \left[ \log p(o_t|s_t) \right]}_{\text{accuracy}},
\end{aligned}$$

where the first expression highlights the evidence bound on the model's evidence, and the second expression shows the balance between the complexity of the state model and the accuracy of the likelihood one. From the latter, the $\mathcal{F}_{\mathrm{AIF}}$ (Equation 2) can be obtained by expliciting $p(s_t)$ as $p(s_t|s_{t-1}, a_{t-1})$, according to the Markov assumption, and by choosing $q(s_t) = q(s_t|o_t)$ as the approximate variational distribution.

## A.2  Free Energy of the Future

For the future, the agent selects actions that it expects to minimize the free energy. In particular, active inference assumes that the future's model of the agent is biased towards its preferred outcomes, distributed according to the prior $\tilde{p}(o_t)$. Thus, we define the agent's generative model as $\tilde{p}(o_t, s_t, a_t) = p(a_t)p(s_t|o_t)\tilde{p}(o_t)$ and we aim to find the distributions of future states and actions by applying variational inference, with the variational distribution $q(s_t, a_t)$. If we consider expectations taken over trajectories sampled from $q(o_t, s_t, a_t) = p(o_t|s_t)q(s_t, a_t)$, the expected free energy $\mathcal{G}$ (Equation 3) becomes:

$$\begin{aligned}
\mathcal{G} &= E_{q(o_t, s_t, a_t)} \left[ \log q(s_t, a_t) - \log \tilde{p}(o_t, s_t, a_t) \right] \\
&= E_{q(o_t, s_t, a_t)} \left[ \log q(a_t|s_t) + \log q(s_t|s_{t-1}, a_{t-1}) - \log p(a_t) - \log p(s_t|o_t) - \log \tilde{p}(o_t) \right],
\end{aligned}$$

where we explicit conditioning on the previous state-action $(s_{t-1}, a_{t-1})$ for the sake of clarity.

We now assume $p(s_t|o_t) \approx q(s_t|o_t)$, which means assuming the variational state posterior model approximates the true posterior over states, as a consequence of minimizing $\mathcal{F}$. Thus, we can rewrite the above result as:

$$\mathcal{G} \approx E_{q(o_t, s_t, a_t)} \left[ \log q(a_t|s_t) + \log q(s_t|s_{t-1}, a_{t-1}) - \log p(a_t) - \log q(s_t|o_t) - \log \tilde{p}(o_t) \right].$$

Then, we assume that the agent's model likelihood over actions is uniform and constant, namely $p(a_t) = \frac{1}{|\mathcal{A}|}$:

$$\mathcal{G} \approx E_{q(o_t, s_t, a_t)} \left[ \log q(a_t|s_t) + \log q(s_t|s_{t-1}, a_{t-1}) - \log q(s_t|o_t) - \log \tilde{p}(o_t) \right] - \log \frac{1}{|\mathcal{A}|}.$$

Finally, by dropping the constant and rewriting all terms as KL divergences and entropies, we obtain:

$$\mathcal{G}_{\text{AIF}} = -E_{q(o_t)}[D_{\text{KL}}\left[q(s_t|o_t)||q(s_t|s_{t-1}, a_{t-1})\right]] - E_{q(o_t)}[\log \tilde{p}(o_t)] - E_{q(s_t)}[\mathcal{H}(q(a_t|s_t))]$$

that is the expected free energy as described in Equation 4.

## B  Model Details

The world model, composed by the prior network $p_\phi(s_t|s_{t-1}, a_{t-1})$, the posterior network $q_\phi(s_t|s_{t-1}, a_{t-1}, o_t)$ and the representation model $f_\phi(o, s)$, is presented in Figure 7.

The prior and the posterior network share a GRU cell, used to remember information from the past. The prior network first combines previous states and actions using a linear layer, then it processes the output with the GRU cell, and finally uses a 2-layer MLP to compute the stochastic state from the hidden state of the GRU. The posterior network also has access to the features computed by a 4-layer CNN over observations. This setup is inspired on the models presented in [21, 54, 4]. For the representation model, on the one hand, we take the features computed from the observations by the posterior's CNN, process them with a 2-layer MLP and apply a $tanh$ non-linearity, obtaining $h_\phi(o)$. On the other hand, we take the state $s_t$, we process it with a 2-layer MLP and apply a $tanh$ non-linearity, obtaining $g_\phi(s)$. Finally, we compute a dot-product, obtaining $f_\phi(o, s) = h_\phi(o)^T g_\phi(s)$. In the world model's loss, $J_\phi = \sum_t \mathcal{F}_{\text{NCE}}(s_t, o_t)$, we clip the KL divergence term in the $\mathcal{F}_{\text{NCE}}$ below 3 free nats, to avoid posterior collapse.

The behavior model is composed by the action network $q_\theta(a_t|s_t)$ and the expected utility network $g_\psi(s_t)$, which are both 3-layer MLPs. In order to get a good estimate of future utility, able to trade off between bias and variance, we used GAE($\lambda$) estimation [43]. In practice this translates into approximating the infinite-horizon utility $\sum_{k=T}^{\infty} \mathcal{G}_{\text{NCE}}(s_t)$ with:

$$G_t^\lambda = \mathcal{G}_{\text{NCE}}(s_t) + \gamma_t \begin{cases} (1 - \lambda)g_\psi(s_{t+1}) + \lambda G_{t+1}^\lambda & \text{if} \quad t < H, \\ g_\psi(s_H) & \text{if} \quad t = H, \end{cases}$$

where $\lambda$ is an hyperparameter and $H$ is the imagination horizon for future trajectories. Given the above definition, we can rewrite the actor network loss as: $J_\theta = \sum_t G_t^\lambda$ and the utility network loss with $J_\psi = \sum_t (g_\psi(s_t) - G_t^\lambda)^2$. In $\mathcal{G}_{\text{NCE}}$, we scale the action entropy by $3 \cdot 10^{-4}$, to prevent entropy maximization from taking over the rest of the objective. In order to stabilize training, when updating the actor network, we use the expected utility network and the world model from the previous epoch of training.

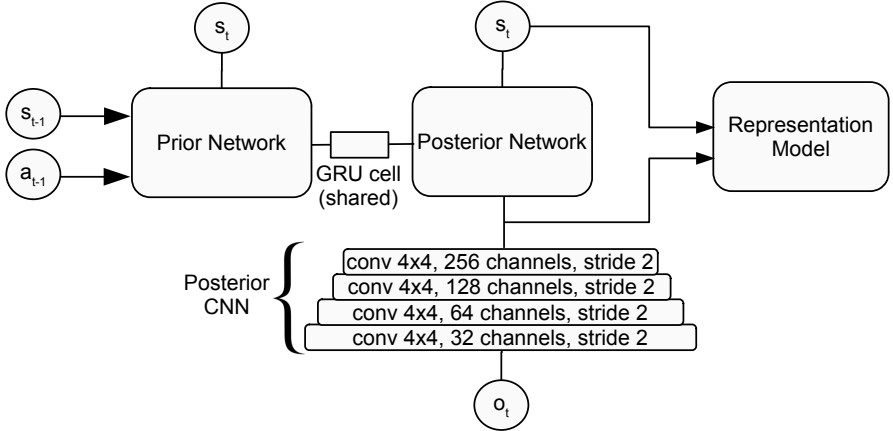

Figure 7: **World Model.** Prior, posterior and representation models. For the posterior CNN, the configuration of each layer is provided.

# C Hyper Parameters

| Name | Value |
|---|---|
| **World Model** | |
| Latent state dimension | 30 |
| GRU cell dimension | 200 |
| Adam learning rate | $6 \cdot 10^{-4}$ |
| **Behavior Model** | |
| $\gamma$ parameter | 0.99 |
| $\lambda$ parameter | 0.95 |
| Adam learning rate | $8 \cdot 10^{-5}$ |
| **Common** | |
| Hidden layers dimension | 200 |
| Gradient clipping | 100 |

Table 3: World and behavior models hyperparameters.

# D Experiment Details

**Hardware.** We ran the experiments on a Titan-X GPU, with an i5-2400 CPU and 16GB of RAM.

**Preferred Outcomes.** For the tasks of our experiments, the preferred outcomes are 64x64x3 images (displayed in Figure 2, 4b, 4a). Corresponding $p(\tilde{o}_t)$ distributions are defined as 64x64x3 multivariate Laplace distributions, centered on the images' pixel values. We also experimented with 64x64x3 multivariate Gaussians with unit variance, obtaining similar results.

**Baselines.** In section 5, we compare four different flavors of model-based control: Dreamer, Contrastive Dreamer, Likelihood-AIF and Contrastive-AIF. Losses for each of these methods are provided in Table 4, adopting the following additional definitions:

$$J_R = -\log p(r_t|s_t)$$
$$G_{\text{RL}} = -E_{q(s_t,a_t)}[r(s_t,a_t)] - E_{q(s_t)}[\mathcal{H}(q(a_t|s_t))],$$

where $G_{\text{RL}}$ is the same as in Equation 5.

| | $J_\phi$ | $J_\theta$ | $J_\psi$ |
|---|---|---|---|
| Dreamer | $\mathcal{F}_{\text{AIF}} + J_R$ | $G_{\text{RL}}$ | $(g_\psi - \sum_t^\infty G_{RL})^2$ |
| Contrastive Dreamer | $\mathcal{F}_{\text{NCE}} + J_R$ | $G_{\text{RL}}$ | $(g_\psi - \sum_t^\infty G_{RL})^2$ |
| Likelihood-AIF | $\mathcal{F}_{\text{AIF}}$ | $\mathcal{G}_{\text{AIF}}$ | $(g_\psi - \sum_t^\infty \mathcal{G}_{AIF})^2$ |
| Contrastive-AIF | $\mathcal{F}_{\text{NCE}}$ | $\mathcal{G}_{\text{NCE}}$ | $(g_\psi - \sum_t^\infty \mathcal{G}_{NCE})^2$ |

Table 4: **Baselines overview.** All losses are summed over multiple timesteps.

**Distracting Suite Reconstructions.** In the Reacher Easy experiment from the Distracting Control Suite, we found that Dreamer, a state-of-the-art algorithm on the DeepMind Control Suite, was not able to succeed. We hypothesized that this was due to the world model spending most of its capacity to predict the complex background, being then unable to capture relevant information about the task.

In Figure 8, we compare ground truth observations and reconstructions from the Dreamer posterior model. As we expected, we found that despite the model correctly stored information about several details of the background, it missed crucial information about the arm pose. Although better world models could alleviate problems like this, we strongly believe that different representation learning approaches, like contrastive learning, provide a better solution to the issue.

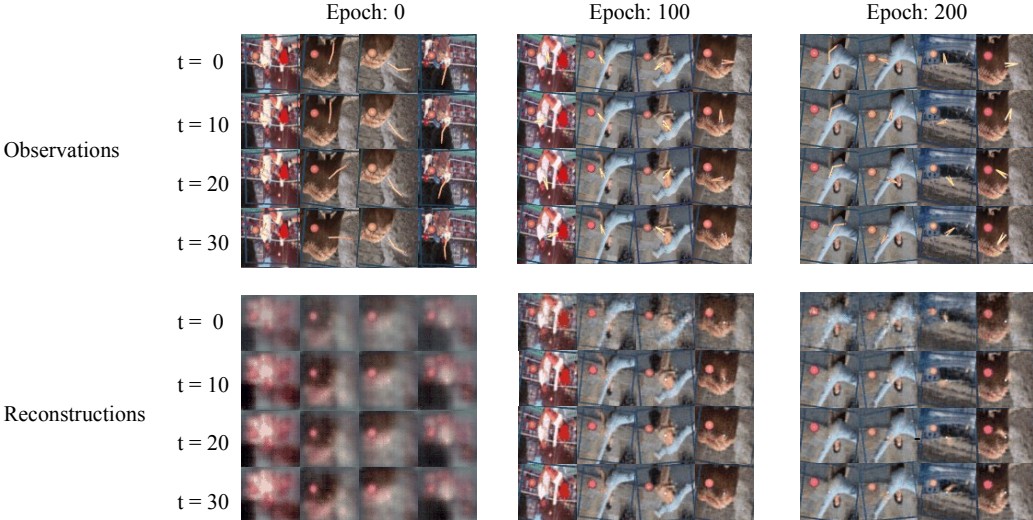

Figure 8: **Dreamer Reconstructions**