# OpenReview forum: "Contrastive Active Inference"
_NeurIPS.cc/2021/Conference — NeurIPS 2021 Poster_

### Official Review · Reviewer_FFRc · 2021-07-08

**Rating:** 7
**Confidence:** 3

**Summary:**

The authors work in the area of active inference, using a contrastive function in optimization to enhance scalability.  They show that their method significantly outperforms non-contrastive active inference on their tasks, and are competitive with some other RL approaches.


**Ethical Concerns:**

I don't see any ethical concerns in this work.


**Limitations And Societal Impact:**

The authors describe limitations of their work in Section 7.

They state in the Checklist that they do not see potential negative impacts, but do not address it in the main body of the manuscript.


**Main Review:**

The modifications made in this paper to standard active inference are not complicated, but they seem novel and effective.  This could be a significant result in making active inference more practicable, and it seems to merit acceptance.

Line 92: "good approximate" -> "good approximation"
Line 109: "that aim" -> "which aim"
Line 146: "as pixel" -> "as pixels"
Line 198: don't capitalize "Section"


**Time Spent Reviewing:**

1

---

> ### Author Response · Authors · 2021-08-09
> **Response to Reviewer FFRc**
>
> We would like to thank the reviewer for their positive feedback and suggestions. We have updated our paper according to their proposed corrections.

---

### Official Review · Reviewer_SRcQ · 2021-07-16

**Rating:** 6
**Confidence:** 2

**Summary:**

This paper proposes a framework for contractive active inference, leveraging contrastive learning to improve active inference in a self-supervised manner. Experiments in goal-conditioned tasks show the proposed algorithm works comparably with reward-based reinforcement learning methods and outperforms the non-contrastive variant of its own. Experiments in an image-based learning task show the proposed method can reach the desired goal despite visual differences in background.

**Limitations And Societal Impact:**

The authors have adequately addressed limitations and potential societal impact.


**Main Review:**


Pros: This paper builds on the idea of active inference, which is a very interesting topic, and proposes to incorporate contrastive learning for improving the efficiency of active inference. Active inference is an important research topic/idea that can potentially address some of reinforcement learning’s shortcomings. The proposed method makes active inference more efficient through leveraging contrastive learning. This paper is well-written with clarity. The experiments demonstrate various advantages of the proposed algorithm.

Cons: The experiments are conducted in two simple domains, gridworlds and a 2D control task. Many goal-conditioned RL algorithms have been successfully applied to high-dimensional control tasks. It is important to also compare the proposed algorithm with RL baselines in these more complex settings and evaluate how it scales.



**Time Spent Reviewing:**

1

---

> ### Author Response · Authors · 2021-08-09
> **Response to Reviewer SRcQ**
>
> Thanks to the reviewer for the effort of reviewing our paper.
>
> > Pros: This paper builds on the idea of active inference, which is a very interesting topic, and proposes to incorporate contrastive learning for improving the efficiency of active inference. Active inference is an important research topic/idea that can potentially address some of reinforcement learning’s shortcomings. The proposed method makes active inference more efficient through leveraging contrastive learning. This paper is well-written with clarity. The experiments demonstrate various advantages of the proposed algorithm.
>
> Thanks!
>
> > Cons: The experiments are conducted in two simple domains, gridworlds and a 2D control task. Many goal-conditioned RL algorithms have been successfully applied to high-dimensional control tasks. It is important to also compare the proposed algorithm with RL baselines in these more complex settings and evaluate how it scales.
>
> We acknowledge that experimenting only in two domains may seem limited and that goal-conditioned behaviour can certainly be a promising application of our work. However, training with multiple goals at once requires several modifications of our algorithm (such as introducing a goal-dependent policy, which provides actions for goal-state pairs of inputs) and more engineered solutions to improve efficiency (such as employing an Hindsight Experience Replay [1]).
>
> We believe that adding complexity on top of a simple model-policy training routine would have further hidden the advantages brought by our proposal. The sets of environments, despite its limitedness, was chosen to look directly into some properties of our method:
> 1. providing a dense objective for behaviour learning, in the absence of rewards (or in sparse rewards settings);
> 2. performing better than likelihood-based methods, when trying to match a goal image, using the contrastive formulation;
> 3. ability to generalize to semantically close goals, despite visual differences.
>
> The experiments we adopted, including sparse-reward tasks and control in presence of complex backgrounds, was explicitly chosen to look into these properties.

---

### Official Review · Reviewer_kRCS · 2021-07-16

**Rating:** 7
**Confidence:** 4

**Summary:**

This work proposes an active inference implementation that scales in high dimensional observations and continuous actions spaces. To achieve this the authors propose to replace the reconstruction required to optimize the free energy objective with the contrastive loss which aims to maximize the mutual information between the state and the observation at each timestep and is more robust under distractors and low capacity models. They show comparable performance to state of the art model-based RL baselines (dreamer, CVRL) and they demonstrate robustness under distractors setting. Also, the provided code is great and that would make a very nice contribution as well (the existing implementations of AIF are too difficult to parse I think)

**Limitations And Societal Impact:**

the authors discuss the broader impact

**Main Review:**

However, my main concern is regarding the experimental setup.

Questions to the authors:

Did you compare with the original implementation of Dreamer and CVRL? If so, which versions did you use? Dreamer v1 or v2? Comparing between dreamer / cvrl / aif / contrastive aif would also introduce confounding effects by the difference in architectures / code if any.

figure 6: what is the x axis? does 100 represent 100*10^5 steps?

Also in the CVRL paper and reacher easy / natural environment CVRL achieves around 990 and dreamer around 133. Is the distracting environment

In active inference, the intrinsic motivation module will provide gradients when the prior and posterior have different values. However, in mujoco tasks, the POMDPness is coming from the fact that you'd need several frames to estimate the true state which is already provided to you via the RNN. As a result, observations won't reduce the uncertainty of the posterior belief significantly and the KL would be mostly constant (I might be totally wrong but my intuition is that mujoco tasks are fairly observable - only velocity/acceleration information is missing which can be inferred with 2-3 frames).

So, the main question is the following: For the selected tasks, the optimal policy doesn't seem to require an information-seeking behaviour and as such the reward would end up being dominated by the extrinsic value, in which case it should provide any additional benefit over dreamer.

I like this work a lot but i think the experimental section doesn't sell it. You'd need experiments that show how AIF is better than dreamer (exploration) and how CAIF is better than AIF (distractors/capacity). But the way is currently designed, the comparison is only between CAIF and {Dreamer / AIF} (that's because the intrinsic value would be mostly zero on those environments).

Misc comments:

Consider referring to the agent as neutral "it" instead of "he" (e.g. line 149)

line 30: "... fulfil optimistic predictions ..." - how's optimistic meant here? consider omitting optimistic for clarity

line 35: ... have replaced the agents' preferred outcomes with RL like rewards ... - How is it different to the extrinsic value of your proposal? In fact, in [13] the authors follow the AIF formulation which is similar to this work. Additionally, they assume the active learning objective which is closer to the original formulation of AIF (they have the KL between prior (eq 7b) and posterior plus the active learning objective (eq 7b))

Contrastive examples: How did you select negative examples? Please state this explicitly in the main text.

Besides those concerns, I really like the code quality and I see great value at releasing it.


**Time Spent Reviewing:**

8

---

> ### Author Response · Authors · 2021-08-09
> **Response to Reviewer kRCS**
>
> Thanks to the reviewer for reviewing our work.
>
> > Did you compare with the original implementation of Dreamer and CVRL? If so, which versions did you use? Dreamer v1 or v2? Comparing between dreamer / cvrl / aif / contrastive aif would also introduce confounding effects by the difference in architectures / code if any.
>
> In order to reduce any confounding effects introduced by different codebases / ML frameworks / architectures, we used the same codebase for all methods. For Dreamer, we faithfully reimplemented DreamerV1 in PyTorch, following the original codebase
> (available here: https://github.com/danijar/dreamer). Starting from that, we implemented all other baselines, with the same architecture and parameters.
>
> At the moment, only the code corresponding to our method is present in the Supplementary Material, to keep things cleaner and easy to follow. However, we are willing to also share the code for all the other baselines, if that could be useful for the community.
>
> > figure 6: what is the x axis? does 100 represent 100*10^5 steps?
>
> In both Figure 3 and Figure 6, the x axis represents the number of trajectories/episodes. We chose this metric as in Minigrid the length of an episode depends on whether the agent successfully achieves the task or not. For the control experiments, the length of an episode is, instead, fixed to 1000 steps, so that 100 episodes is equivalent to 10e5 steps. To clear any confusion, we will add this explanation to the experiments’ section.
>
> > Also in the CVRL paper and reacher easy / natural environment CVRL achieves around 990 and dreamer around 133. Is the distracting environment
>
> We believe part of the comment was lost. We chose to use the Distracting suite instead of the Natural suite in CVRL as it constitutes both a cleaner implementation and a more complex benchmark.
>
> As shown here (https://github.com/Yusufma03/CVRL/blob/d95cc9c98939626d395ea8c150d71344b18e0844/wrappers.py) the Natural suite backgrounds were naively replaced by extracting all yellow pixels in the image and stacking them on top of a different background. The Distracting Control Suite [46], instead, directly modifies the background of the DeepMind Control Suite, leading to cleaner results, and also introduces variations in camera angles and colors, for extra difficulties.
>
> > So, the main question is the following: For the selected tasks, the optimal policy doesn't seem to require an information-seeking behaviour and as such the reward would end up being dominated by the extrinsic value, in which case it should provide any additional benefit over dreamer.
>
> > I like this work a lot but i think the experimental section doesn't sell it. You'd need experiments that show how AIF is better than dreamer (exploration) and how CAIF is better than AIF (distractors/capacity). But the way is currently designed, the comparison is only between CAIF and {Dreamer / AIF} (that's because the intrinsic value would be mostly zero on those environments).
>
> We thank the reviewer for appreciating our work and we agree with them in that the paper does not investigate in what situations Active Inference would provide advantages over RL techniques (e.g. Dreamer). While we believe that in some contexts the ambiguity minimization nature of the AIF objective should indeed lead to better exploration / risk-sensitive behaviours, a comparison of RL and AIF was not in the scope of our work.
>
> The choice of our baselines was driven by the intention of showing that:
> 1. our method facilitates learning compared to likelihood-based AIF, as shown in all the experiments;
> 2. our method can achieve goals provided as observations almost as efficiently as having access to privileged filtered information (rewards), as shown in all experiments;
> 3. likelihood-based models are too sensitive to complex backgrounds/pixel-wise variations in the environment, shown in the experiments on the Distracting Control suite.
>
> Finally, and most surprisingly, our method was able to link the goal represented in the standard DeepMind Control suite to the more complex setup of the Distracting suite, which represents a propensity to generalize that likelihood-based models miss.
>
> > line 35: ... have replaced the agents' preferred outcomes with RL like rewards ... - How is it different to the extrinsic value of your proposal? In fact, in [13] the authors follow the AIF formulation which is similar to this work. Additionally, they assume the active learning objective which is closer to the original formulation of AIF (they have the KL between prior (eq 7b) and posterior plus the active learning objective (eq 7b))
>
> Rewards are a form of privileged information, which can require considerable human engineering efforts of trial and error, as well as leading to unexpected behaviours (some examples in [10,28,37]).
> In some environments, the best way to provide rewards is to give the agent a sparse signal of +1 in the correct state and 0 otherwise. For RL, if no initial exploration is provided, this can be quite a problem, as the expected return quickly becomes zero, leading to ineffective gradients.
>
> Our objective, as well as the standard AIF one (not using rewards), can be seen as a semi-supervised one, where the only knowledge that the agent requires is its preferred observations distribution. In biological agents, preferred outcomes are naturally available, such as avoiding dangers, satisfying the needs of sleeping and eating. In artificial AIF agents, we instill such knowledge by simply defining a distribution centered around the desired outcome(s), which we also believe is a much easier process than engineering rewards. The objective that derives from this formulation is a dense function over the possible outcomes, that we show and compare to rewards in Figure 4, for the MiniGrid task.
>
> > Contrastive examples: How did you select negative examples? Please state this explicitly in the main text.
>
> At each training iteration of the model, we sample B trajectories of length L from the replay buffer D. In this way, for each observation, L-1 negative examples are represented by intra-episode negatives, that are observations that are temporally different, and (B-1)*L are extra-episode negatives, which are observations from different episodes (and at different timesteps).
>
> We will include such a description in the main text, as requested.
>
> > Besides those concerns, I really like the code quality and I see great value at releasing it.
>
> Thanks!!!
>
> We have addressed other misc comments of the reviewer and updated our paper accordingly.

---

> > ### Comment · Reviewer_kRCS · 2021-09-02
> > **thank you**
> >
> > I'd like to thank the authors for addressing my comments. I've increased my score to reflect my satisfaction with their response.

---

### Official Review · Reviewer_WaHE · 2021-07-17

**Rating:** 7
**Confidence:** 4

**Summary:**

This paper introduces a novel variation of the “Active Inference Model” that by incorporating contrastive learning instead of likelihood-based learning, makes the learning more computationally tractable and not reliant on high-dimensional observation reconstriction.

**Limitations And Societal Impact:**

Yes.

**Main Review:**

Finding alternative methods to reward learning is an important research direction that could be fruitful in having more generalizable and biologically plausible models. In addition, recently there have been very promising results in using contrastive learning in a wide array of domains. And this work takes advantage of those recent advances in contrastive learning to push forward the active inference model. The paper is well written and it is a pleasure to read. And the authors do discuss some of the drawbacks of their work.
The main shortcoming of the paper is its experiments section, which is limited to only two simple tasks. Would be interesting to see the performance in a wider array of tasks and more in-depth analysis.


Specifying desired outcomes might pose its own challenges similar to those of reward design. Could you please elaborate more on that?

Could you please describe more in detail how $\tilde{p}(o_t)$ formulated for each task?

Do you have any results or intuition on how the agent might behave when the goal is not fixed in the reacher task?

**Time Spent Reviewing:**

4

---

> ### Author Response · Authors · 2021-08-09
> **Response to Reviewer WaHE**
>
> We thank the reviewer for their useful feedback.
>
> > The paper is well written and it is a pleasure to read.
>
> Thanks!
>
> > (...) Would be interesting to see the performance in a wider array of tasks and more in-depth analysis.
>
> We agree that a more extensive analysis would have strengthened the presentation. Nonetheless, the experiments that we presented were particularly suited to show where our method shines: 1) settings with sparse rewards, where our method provides a denser objective; 2) settings with distractions/variations in the background, where providing a pixel-perfect goal is nearly impossible (as it generally happens in real-world settings). To cover a more extensive set of settings, we also experimented with both discrete and continuous action spaces.
>
> Given our limited compute budget, we had to focus on a limited number of tasks, but evaluating on a wider and more complex array of tasks is definitely something we'll investigate in future work.
>
> > Specifying desired outcomes might pose its own challenges similar to those of reward design. Could you please elaborate more on that?
>
> Reward functions are straightforward in some domains, like video games, where the game scores can be used as rewards (besides not always leading to the expected results [10]). Despite this, we believe that, in more realistic domains, it is easier to think of the goals we want our artificial agents to achieve in terms of observations, rather than as an objective function to maximize (i.e. the reward function). For instance, in robotic manipulation tasks, we generally ask a robot arm to act on the environment in order to achieve a desired configuration, e.g. a set of stacked bricks. Engineering a reward for this behaviour can be problematic, as discussed in [37]. Conversely, assembling the final setup to achieve and showing it to the agent is much more convenient.
>
> In the real world, collecting pixel-perfect observations that represent desired outcomes is certainly a challenge, as many factors keep varying (lighting, presence of distractions, background). So, we agree with the reviewer in that there are some issues with defining desired outcomes in terms of observations. However, while the issue is hard to overcome for likelihood-based representations, contrastive learning (and, potentially, better self-supervised representations in the future) have an easier time with it. This is also one of the reasons why our approach represents a breakthrough towards more applicable Active Inference and goal-achieving behaviour, and our experiments with the Distracting Control suite also present some empirical evidence for this.
>
> > Could you please describe more in detail how $\tilde{p}(o_t)$  formulated for each task?
>
> For the tasks of our experiments, the preferred outcome images are shown in Figure 2, Figure 5a and Figure 5c. The corresponding distributions $\tilde{p}(o_t)$ are defined as 64x64x3 multivariate Laplace distributions, centered on the images’ pixel values. We also experimented with 64x64x3 multivariate Gaussians with unit variance, obtaining similar (actually, slightly worse) results. This detail will be included in the Appendix.
>
> > Do you have any results or intuition on how the agent might behave when the goal is not fixed in the reacher task?
>
> This is a really interesting question. Looking specifically at the Reacher task example, we believe the agent might have a hard time reaching different goals by using only the desired outcomes we presented in Figure 5. The reason why is that the image represents two concepts: 1) the arm standing in a very specific position and 2) the arm traversing the red sphere. What should the arm do if the red sphere moves? Get in the right position or traverse the red sphere? This is ambiguous.
>
> Running a few preliminary experiments, we found that the arm rather tends to imitate the given outcome’s arm position rather than following the sphere. In order to overcome this problem, multiple images of the desired outcome can be provided, so that the common goal becomes less ambiguous. Such more complex scenarios will be part of future studies.

---

### Decision · Program_Chairs · 2021-09-27

**Decision:**

Accept (Poster)

**Comment:**

The paper proposes a method that bypasses the need for reconstruction in MBRL using a contrastive objective. The proposed method outperforms Dreamer on the domains studied in the paper. All reviewers unanimously vote to accept the paper, which I agree with. I would encourage the authors to scale up with their method and present results on at least a few ATARI environments in the camera-ready version.